# MULTILAYERDIFFUSION: COMPOSING GLOBAL CONTEXTS AND LOCAL DETAILS IN IMAGE GENERATION

## ABSTRACT

Diffusion models have demonstrated their capability to synthesize high-quality and diverse images from textual prompts. However, simultaneous control over both global contexts (e.g., object layouts and interactions) and local details (e.g., colors and emotions) still remains a significant challenge. The models often fail to understand complex descriptions involving multiple objects and reflect specified visual attributes to wrong targets or forget to reflect them. This paper presents *MultiLayerDiffusion*, a novel framework which allows simultaneous control over the global contexts and the local details in text-to-image generation without requiring training or fine-tuning. It assigns multiple global and local prompts to corresponding layers and composes them to generate images using pre-trained diffusion models. Our framework enables complex global-local compositions, decomposing intricate prompts into manageable concepts and controlling object details while preserving global contexts. We demonstrate that MultiLayerDiffusion effectively generates complex images that adhere to both user-provided object interactions and object details. We also show its effectiveness not only in image generation but also in image editing.

## 1 INTRODUCTION

Text-to-image generative models have emerged recently and demonstrated their amazing capabilities in synthesizing high-quality and diverse images from text prompts. Diffusion models (Dhariwal & Nichol, 2021; Ho et al., 2020; Nichol & Dhariwal, 2021) are currently one of the state-of-the-art methods and widely used for the image generation. Despite their impressive advances in image generation, lack of control over the generated images is a crucial limitation in deploying them to real-world applications.

To provide further controllability over diffusion models, researchers have put a lot of effort into control of object layouts, object interactions, and composition of objects. Chen et al. (2023) have introduced Training-free layout control which takes a text prompt along with the object layout as an input and control the object position based on a loss between the input layout and attention maps. Bar-Tal et al. (2023) and Zhang et al. (2023) have proposed methods to place an object with specified details on a certain region using segmentation masks and a prompt for each segment. Farshad et al. (2023) and Yang et al. (2022) have leveraged a scene graph as an input and introduced models which generate images from the scene graph to control the object interactions in the generated images. However, these methods cannot control both the global contexts (e.g., object interactions and object layouts) and the local details (e.g., object colors and emotions) simultaneously.

Visual scenes can be represented as wholes and parts in nature (Hinton, 2021), i.e., global contexts and local details. Text-to-image models are required to understand both of them from a textual description called a prompt and correctly reflect them in generated images. While texts are flexible and can handle various expressions, the textual specification is a poor way of describing the precise object layout. Although texts are also a linear structure, visual scenes are inherently graph structures. The explicit connection between the global contexts and the local details in the scene may be lost by describing a complex scene in a single prompt. Moreover, the text-to-image models encode the text descriptions as fixed-size latent vectors, while it is difficult to retain all the details of the scene in the fixed-size vector because complex information needs to be squeezed into it.

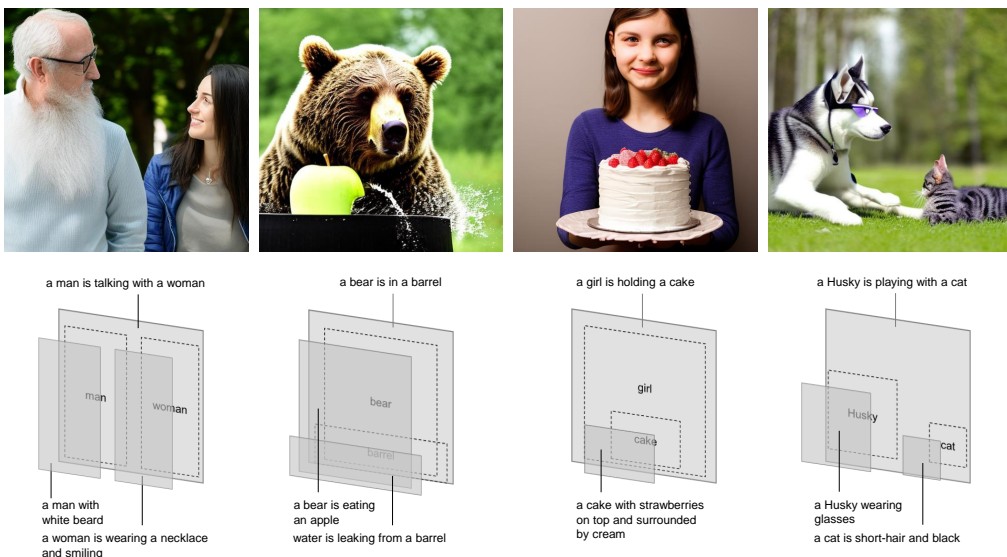

Figure 1: MultiLayerDiffusion takes multiple prompts as an input (e.g., a global prompt: 'a man is talking with a woman' and two local prompts: 'a man with white beard' and 'a woman is wearing a necklace and smiling') along with their layout and assigns noises obtained from them into corresponding layers with a diffusion model. Then, the layers are composed to generate an image. Details of objects in the global layer are guided with the corresponding local layers.

This paper proposes *MultiLayerDiffusion*, a novel diffusion framework that controls both global contexts and local details using a pre-traind diffusion model without requiring any additional training or finetuning. MultiLayerDiffusion takes multiple prompts (e.g., two global prompts and three local prompts) and assigns noises obtained from them into corresponding layers (i.e., a noise from the global prompt is on its global layer and noises from the local prompts are on their local layer) with a diffusion model. We control the object layout on the global layer. Then, the layers are composed to generate an image. Details of objects in the global prompt are guided with the each corresponding local prompt. For example, instead of giving a complex prompt such as 'a man with white beard is talking with a smiling woman wearing a necklace', we decompose it into multiple prompts, a global context 'a man is talking with a woman' and two local details 'a man with white beard' and 'a woman is wearing a necklace and smiling'. The details of the man and the woman in the global prompt are guided with each local prompt in the image generation process. The generated examples are as shown in Fig. 1.

Our framework enables both global-global compositions and global-local compositions. The global-global composition composes foreground and background similar to the existing models, while the global-local composition composes the global context like 'a dog is running with a man' and the details like 'a dog is black' and 'a man is wearing a blue shirt'. Our core contributions are the global-local composition. More than two layers can be composed with those composing methods. Our framework also allows us to control over generated image in both image generation and image editing without finetuning the models.

Our key contributions are summarized as follows[1]:

- We propose MultiLayerDiffusion, a novel framework for diffusion-based image synthesis and editing which enables controlling both global contexts and local details simultaneously.

- Through experimental evaluations, we demonstrate that our proposed method can effectively generate complex images by composing the multi-prompts describing object interactions and details of each object in a scene.

---

[1]The code will be made publicly available upon acceptance.

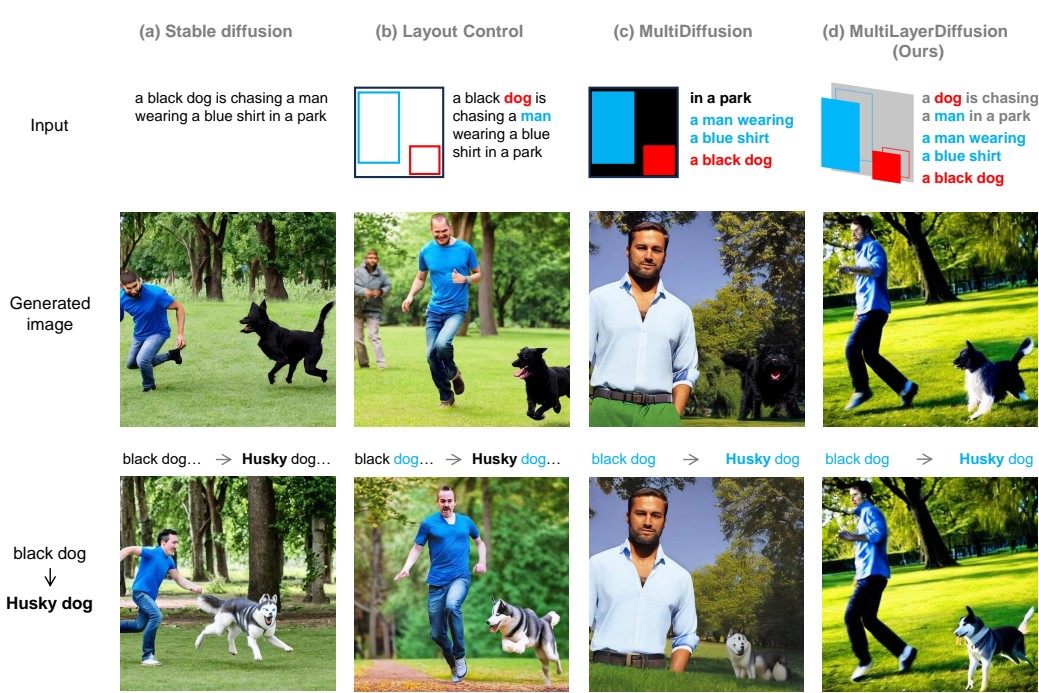

Figure 2: MultiLayerDiffusion enables controlling global contexts (interaction between a dog and a man, their layouts) and local details (the dog is black, the man is wearing a blue shirt) independently. Local details can be specified (black dog → Husky dog) while preserving the global contexts. Note that this is not image editing. We generate images from the text prompts and the layout.

## 2 RELATED WORK

### 2.1 DIFFUSION MODELS

Diffusion models (Dhariwal & Nichol, 2021; Ho et al., 2020; Nichol & Dhariwal, 2021) has attracted a lot of attention as a promising class of generative models that formulates the data generation process as an iterative denoising procedure. The models take a Gaussian noise input $x_T \sim \mathcal{N}(\mathbf{0}, \mathbf{I})$ and transform it into a sample $x_0$ through the series of gradual denoising steps $T$. The sample should be distributed according to a data distribution $q$. Many research works focus on improving the diffusion process to speed up the sampling process while maintaining high sample quality (Nichol & Dhariwal, 2021; Karras et al., 2022). The latent diffusion model (Rombach et al., 2022) has also been developed to address this issue and applied the diffusion process in latent space instead of pixel space to enable an efficient sampling. While the diffusion models have originally shown great performance in image generation, enabling effective image editing and image inpainting (Meng et al., 2022; Avrahami et al., 2023), these models been successfully used in various domains, including video (Ho et al., 2022), audio (Chen et al., 2021), 3D scenes (Müller et al., 2023), and motion sequences (Tevet et al., 2023). Although this paper focuses on image generation, our proposed framework may further be applied in such domains.

### 2.2 CONTROLLABLE IMAGE GENERATION WITH DIFFUSION MODELS

Diffusion models are first applied to text-to-image generative models, which generate an image conditioned on a free-form text description as an input prompt. Classifier-free guidance (Ho & Salimans, 2021) plays important role in conditioning the generated images to the input prompt. Recent text-to-image diffusion models such as DALL-E 2 (Ramesh et al., 2022), Imagen (Saharia et al., 2022), and Stable Diffusion (Rombach et al., 2022) has shown remarkable capabilities in image generation. On the other hand, recent studies (Chen et al., 2023; Bar-Tal et al., 2023; Zheng et al., 2023; Patashnik et al., 2023) have stressed the inherent difficulty in controlling generated images with a text descrip-

tion, especially in the control over (i) object layout and (ii) visual attributes of objects. To gain more control over the object layout, some works have leveraged bounding boxes or segmentation masks as an additional input along with text prompts. Training-free layout control (Chen et al., 2023) takes a single prompt along with a layout of objects appeared in the prompt as shown in Fig. 2 (b). Object layout is given in a form of the bounding. Layout control extracts attention maps from a pre-trained diffusion model and updates the latent embeddings of the image based on an error between the input bounding boxes and the attention maps. Since this method simply uses the pre-trained diffusion model to generate images, it inherits the difficulties in control over the visual attributes of objects, i.e., the local details. Also, the generated image may largely change even if a single word is added or replaced in the prompt due to the fact that the input prompt describes both the global contexts and the local details. MultiDiffusion (Bar-Tal et al., 2023) and SceneComposer (Zeng et al., 2023) take multiple prompts along with their corresponding segmentation masks as a region as shown in Fig. 2 (c). They effectively control the object layout and visual attributes of each object. However, they cannot handle a prompt describing interactions between those objects, i.e., the global contexts. Basically, they just place a specific object described by the prompt in a certain region. Thus, if the input prompt is replaced (black dog $\rightarrow$ Husky dog), the new object (Huskey dog) does not inherit the contexts (e.g., posture) from the replaced object (black dog). Unlike these methods, our method aims to control both the global contexts and the local details simultaneously. Since we treat the global contexts and the local details separately, the global contexts are preserved even if the local details are changed, as shown in Fig. 2 (d). Some studies (Farshad et al., 2023; Yang et al., 2022) focused on image synthesis from scene graphs for better control over the complex relations between multiple objects in the generated images. However, these works require costly extensive training on curated datasets. They regard a complex scene graph as an input prompt, while our approach decomposes the complex prompts into multiple simple prompts and does not require any training or finetuning.

## 2.3 Layered Image Generation and Editing

Some recent works (Zhang et al., 2023; Li et al., 2023; Liao et al., 2023) have proposed layered image generation and editing. They considers two layers, foreground and background, and enables to control them individually with a segmentation mask of the foreground object. Their models are needed to be trained with proposed losses. Unlike them, our goal is to control the global contexts and the local details simultaneously without requiring the training and the accurate segmentation masks as shown in Fig. 1. Our framework enables to control local details while keeping the global contexts by composing the multiple layers, where the global layer may represent the foreground or background and the local layer may represent the details of the objects in the global layer. The global layer and the local layer are not independent but have a whole-part relationship. Our framework can also handle more than two layers as shown in Fig. 7.

## 2.4 Compositional Generation

The compositional generation is an approach to generate the complex images by composing a set of diffusion models, with each of them modeling a certain component of the image. This approach has been an essential direction for image-to-text models because it is difficult for the current models to handle complex prompts where multiple concepts are squeezed. Recently, Liu et al. (2022) has demonstrated successful composition of independent concepts (e.g., "a bear" and "in a forest") by adding estimated score for each concept. Feng et al. (2023) has also proposed another approach which can be directly merged into the cross-attention layers. Inspired by the first approach, we propose a novel method to compose whole-part concepts (e.g., "a bear is eating an apple" and "the apple is green").

## 3 Method

Our goal is to generate images where given global contexts and local details are reflected. In this section, we propose MultiLayerDiffusion to compose the global context and the local detail with pre-trained diffusion models.

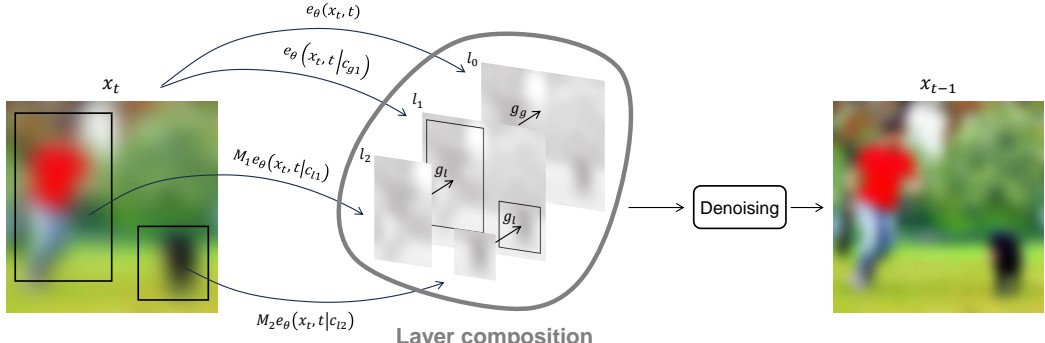

Figure 3: MultiLayerDiffusion composes multiple layers. Unconditional noise and noises conditioned on global contexts (e.g., interactions) or local details (e.g., color) are assigned to separate layers ($l_0$, $l_1$, $l_2$). Those layers are then composed with global guidance $g_g$ and local guidance $g_l$.

## 3.1 COMPOSITIONS OF DIFFUSION MODELS

We consider a pre-trained diffusion model, which takes a text prompt $y \in \mathcal{Y}$ as a condition and generates a intermediate image $\boldsymbol{x}_t \in \mathcal{I} = \mathbb{R}^{H \times W \times C}$:

$$\boldsymbol{x}_{t-1} = \Phi(\boldsymbol{x}_t | y). \tag{1}$$

The diffusion models are also regarded as Denoising Diffusion Probabilistic Models (DDPMs) where generation is modeled as a denoising process. The objective of this model is to remove a noise gradually by predicting the noise at a timestep $t$ given a noisy image $x_t$. To generate a less noisy image, we sample $x_{t-1}$ until it becomes realistic over multiple iterations:

$$\boldsymbol{x}_{t-1} = \boldsymbol{x}_t - \epsilon_\theta(\boldsymbol{x}_t, t) + \mathcal{N}(0, \sigma_t^2 I), \tag{2}$$

where $\epsilon_\theta(\boldsymbol{x}_t, t)$ is the denoising network. Liu et al. (2022) has revealed that the denoising network or score function can be expressed as a compositions of multiple score functions corresponding to an individual condition $c_i$.

$$\hat{\epsilon}(\boldsymbol{x}_t, t) = \epsilon_\theta(\boldsymbol{x}_t, t) + \sum_{i=0}^{n} w_i(\epsilon_\theta(\boldsymbol{x}_t, t|c_i) - \epsilon_\theta(\boldsymbol{x}_t, t)). \tag{3}$$

where $\epsilon_\theta(\boldsymbol{x}_t, t|c_i)$ predicts a noise conditioned on $c_i$ and $\epsilon_\theta(\boldsymbol{x}_t, t)$ outputs an unconditional noise. This equation only focuses on composing individual conditions over entire image, e.g., a foreground condition like 'a boat at the sea' and a background condition like 'a pink sky'.

We regard $\epsilon_\theta(\boldsymbol{x}_t, t|c_i) - \epsilon_\theta(\boldsymbol{x}_t, t)$ as a guidance $g_i$, which guides the unconditional noise toward the noise conditioned on a given condition $c_i$. Then, the composed denoising network is viewed as a compositions of the guidance.

$$\hat{\epsilon}(\boldsymbol{x}_t, t) = \epsilon_\theta(\boldsymbol{x}_t, t) + \sum_{i=0}^{n} w_i g_i. \tag{4}$$

## 3.2 LAYER COMPOSITION

We propose MultiLayerDiffusion to extend the above concept to a composition of a global condition and a local conditions, e.g., a global interaction like 'a man is walking with a dog' and object details like 'the man is smiling' and 'the dog is white'. We consider a set of global conditions $c_g = (c_{g1}, ..., c_{gk})$ and a set of local conditions $c_l = (c_{l1}, ..., c_{lm})$. We also introduce a diffusion

---

**Algorithm 1** MultiLayerDiffusion sampling.

---

**Require:** Diffusion model $\epsilon_\theta(\boldsymbol{x}_t, t)$, global scales $w_i$, local scales $w_j$, global conditions $c_{gi}$, local conditions $c_{lj}$, object region masks $M_j$
1: Initialize sample $\boldsymbol{x}_t \sim \mathcal{N}(\boldsymbol{0}, \boldsymbol{I})$
2: **for** $t = T, \dots, 1$ **do**
3:      $\boldsymbol{x}_t \leftarrow f(\boldsymbol{x}_t, c_{gi}, M)$            ▷ apply layout control $f$ with $c_{gi}$ and $M$
4:      $\epsilon_i \leftarrow \epsilon_\theta(\boldsymbol{x}_t, t | c_{gi})$       ▷ compute conditional scores for each global condition $c_{gi}$
5:      $\epsilon_j \leftarrow \epsilon_\theta(\boldsymbol{x}_t, t | c_{lj})$        ▷ compute conditional scores for each local condition $c_{lj}$
6:      $\epsilon \leftarrow \epsilon_\theta(\boldsymbol{x}_t, t)$              ▷ compute unconditional score
7:      $g_g \leftarrow \sum_{i=0}^{k} w_i(\epsilon_i - \epsilon).$          ▷ compute global guidance with Eq. 5
8:      $g_l \leftarrow \sum_{j=0}^{m} w_j M_j(\epsilon_j - \epsilon_j^b).$       ▷ compute local guidance with Eq. 6
9:      $\boldsymbol{x}_{t-1} \sim \mathcal{N}(\boldsymbol{x}_t - (\epsilon + g_g + g_l), \sigma_t^2 I)$          ▷ sampling
10: **end for**

---

layer $l = (l_0, ..., l_t)$, where a single global condition or multiple local conditions can be assigned to a layer as shown in Fig. 3. Each layer contains a condition at least. We compose the layers with two ways of guidance: (i) *global guidance*, which guides the image with global conditions.

$$g_g = \epsilon_\theta(\boldsymbol{x}_t, t | c_{gi}) - \epsilon_\theta(\boldsymbol{x}_t, t), \tag{5}$$

where the unconditional noise is assigned to a global layer as a base layer and the global noise on the other global layer is composed to the base layer. This is also well known as Classifier-free guidance (Ho & Salimans, 2021). The classifier-free guidance works well on global conditions, while it does not work effectively when we compose the global condition and the local condition since their conditions have some overlap. Thus, we newly propose (ii) *local guidance*, which guides an object on the base layer $b$ conditioned on a condition $c_i^b$ with a local condition $c_{lj}^n$ on layer $n$.

$$g_l = M_j(\epsilon_\theta(\boldsymbol{x}_t, t | c_{lj}^n) - \epsilon_\theta(\boldsymbol{x}_t, t | c_i^b)), \tag{6}$$

where $M_j \subset \{0, 1\}^{H \times W}$ is a region mask of $j$-th region corresponding to the condition $c_{lj}^n$. The local guidance emphasizes the difference between the global condition (e.g., 'a dog') and the local condition (e.g., 'the dog is black'). We use the local guidance to compose the local layers and the global layers, e.g., composing the local noise on $l_2$ and the global noise on $l_1$ as shown in Fig. 3.

### 3.3 LAYOUT CONTROL

Objects may appear somewhere in a generated image without any layout control. To effectively compose the global noise and the local noise, we use Training-free layout control (Chen et al., 2023). More specifically, we use the backward guidance to control the object layout in the global layer before computing the global noise.

Algorithm 1 provides the pseudo-code for composing diffusion noises with the global guidance and the local guidance. Our method composes noises obtained with pre-trained diffusion models during inference without any additional training or finetuning.

## 4 RESULTS

### 4.1 IMPLEMENTATION DETAILS

We evaluate our method on the following conditions. In all experiments, we used Stable Diffusion (Rombach et al., 2022) as our diffusion model, where the diffusion process is defined over a latent space $I = R^{64 \times 64 \times 4}$, and a decoder is trained to reconstruct natural images in higher resolution $[0, 1]^{512 \times 512 \times 3}$. We use the public implementation of Stable Diffusion by HuggingFace, specifically the Stable Diffusion v2.1 trained on the LAION-5B dataset (Schuhmann et al., 2022) as the pre-trained image generation model. We also set Euler Discrete Scheduler (Karras et al., 2022) as the

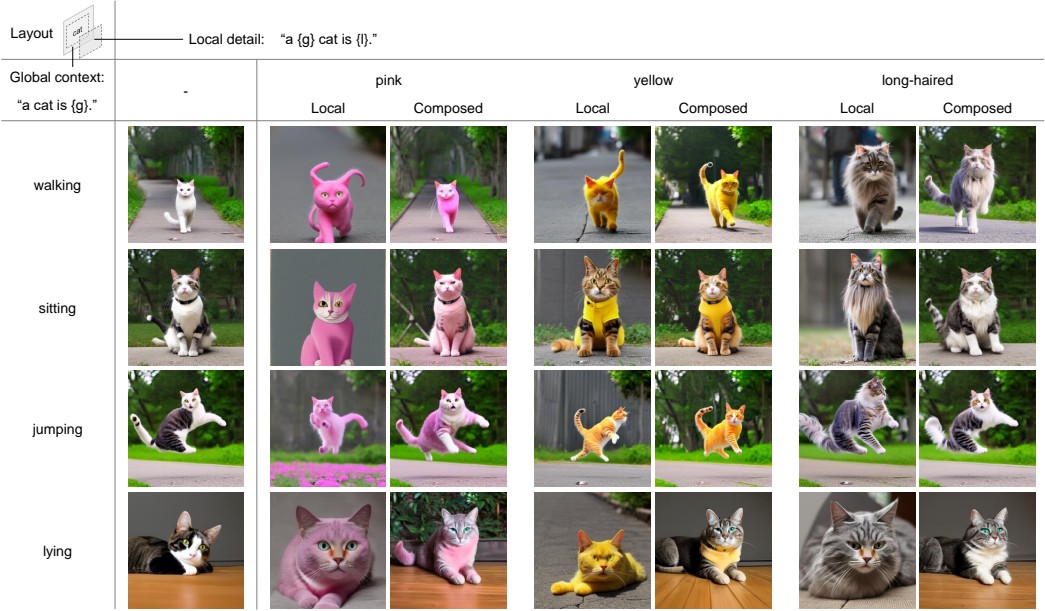

Figure 4: MultiLayerDiffusion for a single object. The images in the first column and 'Local' columns are sampled only from the global context (global images) and the local detail (local images) as an input prompt, respectively. The images in 'Composed' columns are sampled using our method, which effectively applies local detail (e.g., long-haired) to the object in the image while preserving the global contexts (i.e., object layouts and object postures).

noise scheduler. As our layout control (see 3.3) we use the backward guidance proposed in Training-free layout control (Chen et al., 2023). All the experiments are running on one A30 GPU.

We compare our MultiLayerDiffusion with other state-of-the-art training-free methods, including Training-free layout control (Chen et al., 2023) and MultiDiffusion (Bar-Tal et al., 2023), and strong baselines, including OpenAI DALL-E 2 (Ramesh et al., 2022) and Adobe Firefly (Adobe, 2023). We use the publicly available official codes and websites, and follow their instructions.

## 4.2 IMAGE GENERATION WITH MULTILAYERDIFFUSION

We first demonstrate global-local compositions for a single object using MultiLayerDiffusion for a better understanding, while our main targets are more complex scenes including multiple objects as shown in Fig. 1. In Fig. 4, our method generates diverse samples which comply with compositions of a global context (e.g., a cat is walking) and a local detail (e.g., a walking cat is pink). The images in the first column and 'Local' columns are sampled only from the global context (global images) and the local detail (local images) as an input prompt, respectively. The images in 'Composed' columns are sampled using our method, where the goal is to apply local detail (i.e., specified visual attribute) to the object in the image while preserving the global contexts (i.e., object layouts and object postures). Although we generated all the images with the same seed, the posture of the cat is largely different in the corresponding global image and local image (e.g., the image of 'a cat is sitting' vs the image of 'a sitting cat is pink'). Our method can effectively generate the images from the global context and the local detail along with the layout (see 'Composed'). Note that this is image generation not the editing. The generated image retains most of the global contexts, including postures and head directions. In a few cases, the visual attribute of the object changes only partially (e.g., the image composed of 'a cat is lying' and 'a lying cat is yellow').

We then compare MultiLayerDiffusion with the other state-of-the-art methods in generating more complex scene including multiple objects. In Fig. 5, we try to generate images of a complex scene where there are multiple objects in the same category (sheep in this case) and each of them has different attributes (a sheep is black and standing, another sheep is white and sitting). We show

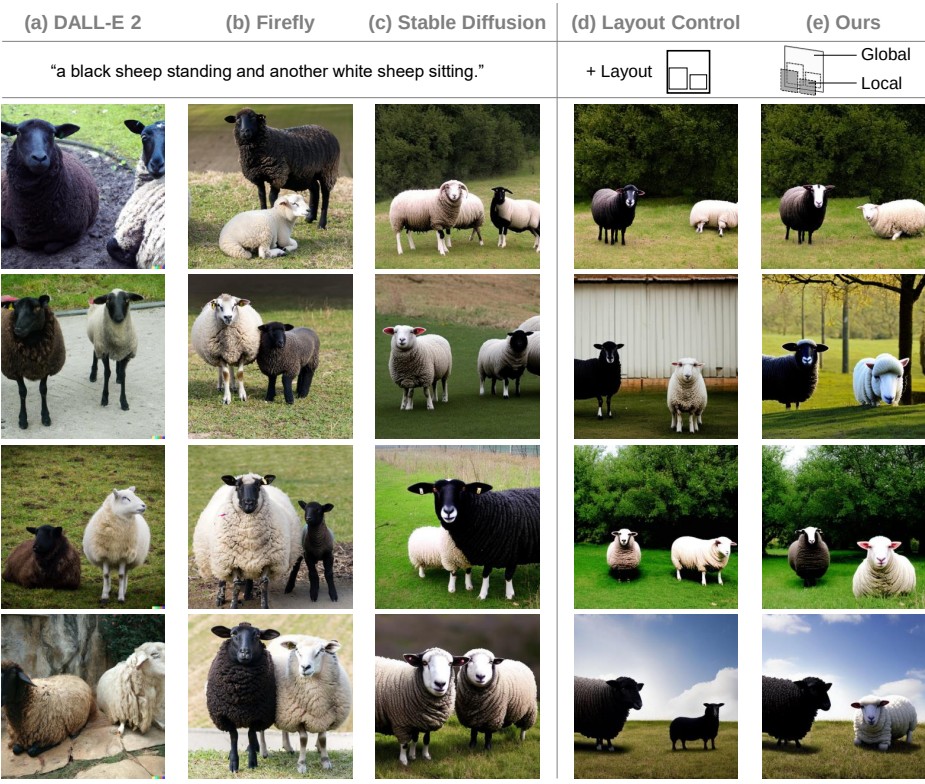

Figure 5: MultiLayerDiffusion for multiple objects. Our method (e) can control attributes of each sheep, while the other methods fail to reflect the specified attributes to the correct targets.

first four images generated by official web application of (a) DALL-E 2 and (b) Firefly at first and second column, respectively. They can generate high-quality images, but they often fail to reflect the specified attributes to the correct targets. We then find four seeds which generate failure samples of Training-free layout control and generate images using Stable diffusion and our method with those seeds. Both the layout control and our method use the same object layout as an additional input. We set 'a black sheep and another white sheep' as a global context, 'a sheep is black and standing' as a local detail of a sheep, and 'a sheep is white and sitting' as a local detail of the another sheep. We make the global context similar to the original prompt to compare the generated images easily. Figure 5 shows that our method effectively controls the attributes of each object in the image.

Figure 1 shows other samples depicting more complex scenes, where we give an interaction between the objects as a global context (e.g., 'a Husky is playing with a cat') and also specify the local details (e.g., 'a Husky wearing glasses' and 'a cat is short-hair and black'). Instead of giving a complex prompt such as 'a Husky wearing glasses is playing with a black short-hair cat', we decompose and handle them separately to effectively synthesis complex visual scenes. MultiDiffusion (Bar-Tal et al., 2023) cannot consider the global context since it just places a conditioned object on a certain region as shown in Fig. 2.

## 4.3 GLOBAL-LOCAL COMPOSITION

We compare our global-local composition with a conventional composition (Liu et al., 2022). Since the conventional composition aims to compose independent concepts (e.g., foreground and background) by adding estimated score for each concept, it often fails to compose overlapped concepts (e.g., 'running cat' and 'white cat') as shown in Fig. 6 (top). Our layer composition can effectively compose such overlapped concepts as shown in Fig. 6 (bottom).

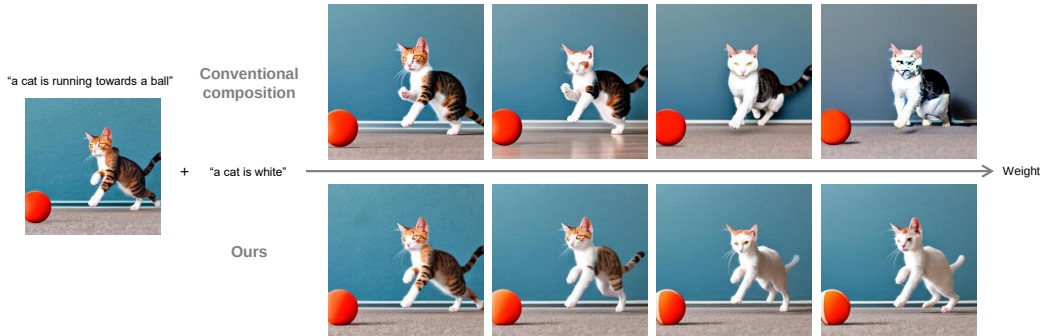

Figure 6: Comparison between our global-local composition (bottom) and the conventional composition (top). Our method can change the detail of the object while preserving the global context by composing two prompts, whereas the conventional method often fails because they regard the prompts as two independent concepts.

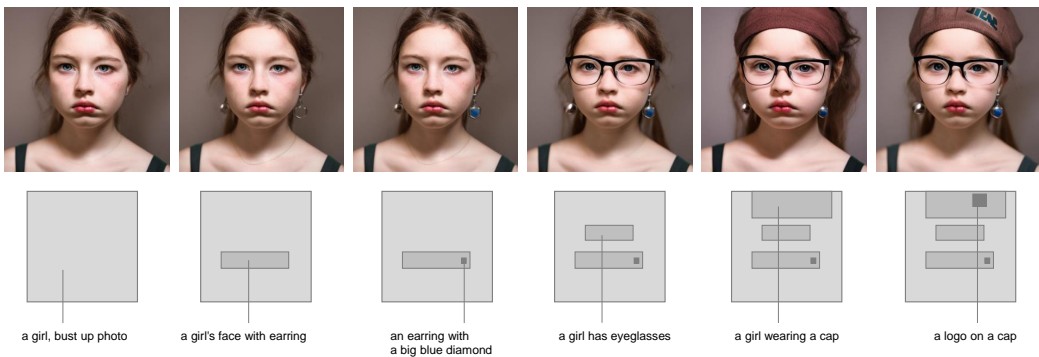

Figure 7: MultiLayerDiffusion also enables layered image editing, where new objects can be added on a certain region using additional prompts. Final image (right end) can be generated in one inference by composing six prompts.

## 4.4 IMAGE EDITING WITH MULTILAYERDIFFUSION

Figure 7 shows edited image samples with MultiLayerDiffusion. MultiLayerDiffusion enables layered image editing, where new objects can be added on a certain region using additional prompts. Details of the objects on a base layer (e.g., earring) can be guided with the prompts on the upper layer (e.g., adding a diamond) with our layer composition. Final image (right end) can be generated in one inference by composing multiple prompts (six in this case).

## 5 CONCLUSION

Image generation with control over both object interactions and each object details is still an open challenge. We proposed MultiLayerDiffusion, a first framework which composes the global context and local detail with a pre-trained diffusion model. Our framework can handle both global-global composition (e.g., foreground and background) and global-local composition (e.g., object layout and object details). Through experimental evaluations, we demonstrated that MultiLayerDiffusion effectively generates images that include interactions between objects with detailed visual control. Unlike existing layered image generation or editing methods, our method can be used for both generation and editing without requiring any training or fine-tuning. A limitation we found is that the object appearance may change only partially when the latent of the object is significantly different between the global layer and the local layer.

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
