# OpenReview forum: "MultiLayerDiffusion: Composing Global Contexts and Local Details in Image Generation"
_ICLR.cc/2024/Conference — ICLR 2024 Conference Withdrawn Submission_

### Official Review · Reviewer_7GYr · 2023-10-18

**Soundness:** 2 fair
**Presentation:** 1 poor
**Contribution:** 2 fair
**Rating:** 3
**Confidence:** 5

**Summary:**

This paper offers separate control over global and local contexts through the utilization of user-specified region masks and prompts for various conditions. This algorithm facilitates precise guidance for image generation and enables localized region editing while preserving the integrity of other regions within the image.

**Strengths:**

This method empowers precise control over both global and local regions, providing a promising avenue for controlled image generation.

**Weaknesses:**

1. The proposed algorithm seems to be an incremental work on MultiDiffusion and Training-free layout control, while primarily relying on Training-free layout control. The key distinction between the proposed algorithm and the aforementioned methods remains somewhat unclear. Is the primary differentiation the replacement of global condition with local guidance in instances where local guidance is employed?

2. The method section presents challenges in terms of comprehension, largely due to the excessive use of undefined notations. For instance:
   - In the Algorithm:
     1) What does the symbol "$f$" represent in Line 3?
     2) How should we interpret "$\epsilon_j^b$" in Line 8?
   - In Section 3.2:
     1) Could you provide a more detailed explanation of the introduced diffusion layer? A mathematical definition or implementation code would be beneficial.
     2) Equation (6) contains numerous subscripts, including "$j$", "$i$", "$b$", and "$n$", while the result introduces a new subscript "$l$". Guidance on how to select "$j$", "$i$", "$b$", "$n$" for a specific "$l$" would be appreciated.

3. Could you provide insights into the methodology for selecting global scales and local scales? This appears to be a pivotal component in combining the various forms of guidance.

4. The experimental section is notably incomplete. There is a lack of quantitative comparisons with competing methods, and an ablation study is conspicuously absent. These omissions limit the comprehensive evaluation of the proposed approach's performance.

**Questions:**

See Weaknesses part.

---

> ### Author Response · Authors · 2023-11-19
>
> Thanks for the review.
>
> **Notations**
> $f$ represents the layout control, which updates the latent variables.
> $e^{b}_{j}$ is the score function corresponding to the condition $c^b_i$, which is the global context of a local detail $c^n_j$.
>
>
> **Novelty**
> Existing methods including MultiDIffusion and Composable diffusion, utilize classifier-free guidance. This approach interprets the distinction between conditional noise and unconditional noise (derived from blank text) as global guidance. In our method, we consider 'dog' in a global prompt as an unconditional dog and 'black long-haired dog' in a local prompt as a conditional dog. Then, we compute the difference between them as local guidance.
>
> We are currently conducting additional evaluations, including quantitative comparisons. See the general response above for our answer.

---

> ### Comment · Reviewer_7GYr · 2023-11-22
> **Thanks for the response**
>
> After reviewing the author's responses, I maintain my original decision. I recommend that the authors revise the paper to enhance clarity and comprehension.

---

### Official Review · Reviewer_m9tU · 2023-10-30

**Soundness:** 2 fair
**Presentation:** 2 fair
**Contribution:** 2 fair
**Rating:** 3
**Confidence:** 5

**Summary:**

The method proposes a training free sampling method to compose both local details (i.e., object attributes) and global context such as a text prompt that describes the general scene using text-to-image diffusion models.

**Strengths:**

- the paper is easy to understand.

**Weaknesses:**

- The writing quality is subpar and needs to be further polished. For example, all quotes are using the wrong quotation marks. The writing needs to further improve and a lot of redundant context and information.
- The experiment is quite limited, where only qualitative results are provided in the main paper. Thus, the performance of method is quite unclear.
- The method novelty is limited, since it simply combines techniques proposed in composable diffusion [1] and lay-out control methods [2] for better compositional generation. I don't see any insightful contributions provided by the paper.

[1] Liu et al., Compositional Visual Generation with Composable Diffusion Models. ECCV 2022 \
[2] Chen et al., Training-free layout control with cross-attention guidance. CVPR 2023

**Questions:**

- I think its worth providing quantitative metrics to showcase the method's performance.
- I do think this paper needs much more efforts to polish and run extensive experiments and ideally provide contributions.

---

> ### Author Response · Authors · 2023-11-19
> **We thank your comments**
>
> Thanks for the review.
>
> **Novelty**
> Our approach does not merely combine layout control [1] and composable diffusion [2] in a straightforward manner. Figure 6 illustrates the comparison between this simple combination and our method. Our approach excels in effectively integrating the global context and local details.
>
> We are currently conducting additional evaluations, including quantitative comparisons. See the general response above for our answer.

---

### Official Review · Reviewer_h5z1 · 2023-11-01

**Soundness:** 2 fair
**Presentation:** 3 good
**Contribution:** 2 fair
**Rating:** 5
**Confidence:** 4

**Summary:**

The paper presents a new framework for simultaneous control over the global contexts and the local details in T2I without requiring additional training or fine-tuning. The key idea of the paper in attaining  detailed visual control is to decompose the complex prompts into manageable concepts and controlling object details while preserving global contexts. Experiments demonstrate the utility of the proposed approach in both generation and editing settings.

**Strengths:**

The proposed idea of layered generation, i.e. decomposing prompts and then guiding the global prompt with each local prompt in the image generation process is very interesting. This is more attractive from practical (e.g. industrial) use cases and is potentially widely applicable as the proposed method doesn’t require additional training and the accurate segmentation masks.

**Weaknesses:**

Although the proposed method is interesting, I feel the paper has few important weaknesses: Firstly, I couldn’t find any quantitative evaluations in the paper. I checked the supplementary and couldn’t find it there either. It is hard to understand the utility of the proposed method to a wide range of prompts without such a comparison. For example, I couldn’t find strong evidence as to why LayoutGPT style methods are not better than the proposed approach. Secondly, it feels to me that the results (layers) with the proposed approach, might not fuse together well in the final generation. It would be helpful to see more qualitative results (supplementary demo shows a very limited samples and the composition looks artificial in those outputs - potentially limiting the applicability of the proposed approach in diverse use cases).

**Questions:**

Please see weaknesses

---

> ### Author Response · Authors · 2023-11-19
> **We thank your comments**
>
> Thanks for the review.
>
> **Novelty**
> LayoutGPT utilizes GLIGEN which is trained to control object layouts, while our method can employ any pre-trained diffusion models, making it widely applicable to various visual domains, such as animation.
>
> We are currently conducting additional evaluations, including quantitative comparisons. See the general response above for our answer.

---

### Author Response · Authors · 2023-11-19
**To all: Thanks for the comments**

We thank the reviewers for their feedback, as it was helpful to improve the paper.

We are currently working on quantitative evaluations to strengthen our study by building a new test set. This task poses a challenge as there is currently no existing test set specifically designed to assess controllability over both global context and local details, as opposed to merely evaluating image quality and color binding.